# Influence of Ultrasound Stimulation on the Viability, Proliferation and Protein Expression of Osteoblasts and Periodontal Ligament Fibroblasts

**DOI:** 10.3390/biomedicines12020361

**Published:** 2024-02-03

**Authors:** Selma Pascoal, Sofia Oliveira, Francisca Monteiro, Jorge Padrão, Rita Costa, Andrea Zille, Susana O. Catarino, Filipe S. Silva, Teresa Pinho, Óscar Carvalho

**Affiliations:** 1UNIPRO—Oral Pathology and Rehabilitation Research Unit, University Institute of Health Sciences (IUCS), CESPU, 4585-116 Gandra, Portugal; selma.pascoal@iucs.cespu.pt (S.P.);; 2Center for MicroElectroMechanical Systems (CMEMS), University of Minho, Campus Azurém, 4800-058 Guimarães, Portugalscatarino@dei.uminho.pt (S.O.C.); oscar.carvalho@dem.uminho.pt (Ó.C.); 3ICVS/3B’s—PT Government Associate Laboratory, 4710-057 Braga, Portugal; 4Center for Textile Science and Technology (2C2T), Department of Textile Engineering, University of Minho, Campus of Azurém, 4800-058 Guimarães, Portugal; padraoj@2c2t.uminho.pt (J.P.); azille@2c2t.uminho.pt (A.Z.); 5LABBELS—Associate Laboratory, 4710-057 Braga, Portugal; 6IBMC—Instituto Biologia Molecular e Celular, i3S—Inst. Inovação e Investigação em Saúde, Universidade do Porto, 4200-135 Porto, Portugal

**Keywords:** fibroblasts, orthodontic tooth movement, osteoblasts, ultrasound

## Abstract

Among the adjunctive procedures to accelerate orthodontic tooth movement (OTM), ultrasound (US) is a nonsurgical form of mechanical stimulus that has been explored as an alternative to the currently available treatments. This study aimed to clarify the role of US in OTM by exploring different stimulation parameters and their effects on the biological responses of cells involved in OTM. Human fetal osteoblasts and periodontal ligament fibroblasts cell lines were stimulated with US at 1.0 and 1.5 MHz central frequencies and power densities of 30 and 60 mW/cm^2^ in continuous mode for 5 and 10 min. Cellular proliferation, metabolic activity and protein expression were analyzed. The US parameters that significantly improved the metabolic activity were 1.0 MHz at 30 mW/cm^2^ for 5 min and 1.0 MHz at 60 mW/cm^2^ for 5 and 10 min for osteoblasts; and 1.0 MHz at 30 mW/cm^2^ for 5 min and 1.5 MHz at 60 mW/cm^2^ for 5 and 10 min for fibroblasts. By stimulating with these parameters, the expression of alkaline phosphatase was maintained, while osteoprotegerin synthesis was induced after three days of US stimulation. The US stimulation improved the biological activity of both osteoblasts and periodontal ligament fibroblasts, inducing their osteogenic differentiation.

## 1. Introduction

During orthodontic tooth movement (OTM), several mechanical and biological processes take place, comprising the remodeling of the periodontal ligament (PDL) and alveolar bone. Remodeling and modeling involve coordinated action of osteoclasts and osteoblasts in response to mechanical load. Moreover, inflammatory mediators, namely, interleukin (IL)-1, IL-2, IL-6, IL-8 and tumor necrosis factor-alpha, are released after a mechanical stimulus or injury, triggering the biologic process associated with OTM [1,2]. Orthodontic treatment is usually considered a long-term treatment, which can lead to several complications, such as a high risk of caries, root resorption and decreased patient compliance and satisfaction. Thus, accelerating OTM and shortening treatment duration is the primary goal of the orthodontist to prevent the detrimental effects of longer treatment times and to meet patients’ expectations [3,4].

In recent years, numerous adjunctive procedures based on surgical (e.g., corticotomy, piezocision) [5,6] and nonsurgical (e.g., photobiomodulation, mechanical stimulation) [7,8] procedures have been introduced to accelerate OTM. Particularly, mechanical stimulation has gained more attention since it has been reported to activate osteoblasts and osteoclasts, promoting bone remodeling [9]. Current in vivo and in vitro studies have applied different mechanical loading regimes and have shown that this stimulus has a beneficial effect on the proliferation and differentiation of bone cells [10,11,12,13,14].

Ultrasound (US) is a form of mechanical stimulus that has been explored as an alternative to the currently available treatments to accelerate OTM. It is transmitted to the human body as high-frequency acoustic pressure waves and it can cause biochemical and biophysical events at the cellular level [15], also promoting bone formation [16,17,18]. This therapy is commonly applied with central frequencies of 1.0 and 1.5 MHz and with an intensity (i.e., power density) of 30 mW/cm^2^ in pulsed mode for 10, 15 or 20 min [19,20,21,22]. There is a lack of studies investigating other parameters that could also be employed during US stimulation. Also, the previous studies [8,23,24] have shown inconsistent results on the efficacy of therapeutic US in orthodontics, which may arise from the use of different stimulation protocols, experimental models and/or tooth movement mechanics.

More in vitro studies should be conducted to better clarify the role of US in OTM by exploring different stimulation parameters and their effects on the biological responses of cells involved in OTM. This study aims to fill this gap in the literature by stimulating human osteoblasts and periodontal ligament fibroblasts with US at 1.0 and 1.5 MHz central frequencies and power densities of 30 and 60 mW/cm^2^, in continuous mode, for 5 and 10 min. The effects of such US parameters were analyzed in terms of proliferation, metabolic activity and protein expression of osteoblasts and fibroblasts involved in the OTM.

## 2. Materials and Methods

### 2.1. Cell Culture

A human fetal osteoblast cell line (hFOBs; American Type Culture Collection—ATCC^®^, Manassas, VA, USA) was cultured in Dulbecco’s Modified Eagle Medium/Nutrient Mixture F-12 (DMEM-F12), without phenol red (PAN-Biotech GmbH, Aidenbach, Germany), containing 2.5 mM L-glutamine (PanBiotech^®^, Aidenbach, Germany), 10% (*v*/*v*) fetal bovine serum (FBS; Sigma-Aldrich, St. Louis, MO, USA) and 0.3 mg/mL geneticin (G418; PAN-Biotech GmbH, Aidenbach, Germany). The hFOBs were used between 12 and 14 passages.

Human periodontal ligament fibroblasts (hPLFs) were isolated from healthy human periodontal tissue and purchased from Innoprot^®^, Spain. They were cultured in DMEM-F12 culture medium supplemented with stable glutamine and 1.2 g/L NaHCO_3_ (PAN-Biotech GmbH, Aidenbach, Germany) with 10% FBS (Sigma-Aldrich) and 1% (*v*/*v*) penicillin–streptomycin (PAN-Biotech GmbH, Aidenbach, Germany). The hPLF cells were used at the third passage. Both hFOBs and hPLFs were maintained at 37 °C in a humidified atmosphere of 5% CO_2_, and the culture medium was changed twice a week.

### 2.2. Cell Seeding

At a 90% confluence, hFOBs and hPLFs were detached from the culture flasks with 0.25% (*v*/*v*) trypsin/EDTA (PAN-Biotech GmbH, Aidenbach, Germany). The hFOBs were seeded at a concentration of 20,000 cells/well and the hPLFs at 30,000 cells/well in 12-well plates for 72 h prior the US stimulation. The 12-well plates were maintained at 37 °C and 5% CO_2_.

### 2.3. Ultrasound Stimulation

A customized device was developed to deliver tailored US stimulation to cells. The customized equipment was developed to stimulate cultured cells in 12-well plates. It was connected to a DC power supply (Uni-Trend Technology Co., Ltd., Shanghai, China) and to a computer with software that allowed for control of the central frequency, power density and duration of stimulation. The ultrasonic transducers consisted of one piezoelectric transducer positioned on top of each well. They were immersed in the culture medium to guarantee the propagation of the acoustic waves (Figure 1).

Untreated cells were used as controls and operated under the same conditions, but with the US equipment turned off. The transducers were disinfected with 70% (*v*/*v*) ethanol and sterilized with UV radiation for 20 min inside a biological safety level 2 vertical laminar airflow cabinet (Gelaire^®^, Seven Hills, Australia). The stimulation was performed in a sterile environment inside the laminar flow cabinet at room temperature, and, following stimulation, the 12-well plates were maintained inside the incubator at 37 °C in a humidified 5% CO_2_ atmosphere.

The US stimulation was performed in two phases to evaluate its effect on cellular behavior. Firstly, hFOBs and hPLFs were stimulated only once at 1.0 and 1.5 MHz with 30 mW/cm^2^ and 60 mW/cm^2^ for 5 and 10 min in continuous mode, and the metabolic activity was analyzed after 1, 24 and 72 h of US stimulation. The stimulation conditions that induced the highest metabolic activity were selected and applied daily for three days to assess the effects of several US stimulation sessions on cellular proliferation, alkaline phosphatase (ALP), osteoprotegerin (OPG) and receptor activator of nuclear factor-kappa β (RANKL) expressions. Four replicas were used for each experimental condition.

### 2.4. Metabolic Activity

For the initial screening, an MTT cell proliferation assay (Celltiter 96^®^, Promega^®^, Madison, WI, USA) was used according to the manufacturer’s indications to assess the metabolic activity. This assay is based on the cellular conversion of a tetrazolium salt into a formazan product that is solubilized and then detected using a 96-well plate reader. After 1, 24 and 72 h of the single US stimulation, hFOBs and hPLFs were incubated with MTT reagent for 4 h at 37 °C in a humidified atmosphere of 5% CO_2_. Afterwards, the solubilization solution was added, and the absorbance was read at 570 nm in a microplate reader (Epoch, BioTek, Santa Clara, CA, USA). To exclude the effect of culture medium, a blank was prepared with only culture medium without cultured cells, and its absorbance was subtracted to the absorbance from the cultured wells.

For the second phase, MTS [3-(4,5-dimethylthiazol-2-yl)-5-(3-carboxymethonyphenol)-2-(4-sulfophenyl)-2H-tetrazolium inner salt (MTS) reagent (Biovision^®^, San Francisco, CA, USA) was used to quantify the metabolic activity. After one, two and three days of daily US stimulation, cells were incubated with MTS reagent for 4 h at 37 °C, 5% CO_2_ in a humidified atmosphere and the absorbance was read at 490 nm using a microplate reader (Epoch, BioTek, Santa Clara, CA, USA). Similarly, a blank was prepared, and the calculations were performed as previously mentioned.

### 2.5. Cell Concentration

After one, two and three days of daily stimulation, hFOBs and hPLFs were trypsinized and counted with 0.4% (*v*/*v*) trypan blue solution (PAN Biotech GmbH, Aidenbach, Germany) using a hemocytometer. Trypan blue is used to count cells based on the fact that viable cells do not take up trypan blue, while the dead cells are permeable and incorporate the dye. The cellular morphology was observed daily with an inverted microscope (Kern^®^, Albstadt, Germany) to check for morphology alterations due to US stimulation.

### 2.6. Alkaline Phosphatase Expression

The ALP activity was measured using a commercial colorimetric kit (BioVision^®^, San Francisco, CA, USA) after one, two and three days of daily US stimulation. For this, culture medium was collected from each condition after each timepoint, and the assay was conducted according to the manufacturer’s protocol. Briefly, 80 μL of culture medium was added to 96-well plates with 5 mM p-nitrophenyl phosphate solution and incubated for 60 min at room temperature, protected from light. Afterwards, stop solution was added to each condition, and the absorbance was measured at 405 nm using a microplate reader (Epoch, BioTek, Santa Clara, CA, USA). A standard curve was prepared, and the amount of ALP in each experimental condition was calculated according to the manufacturer’s protocol.

### 2.7. Osteoprotegerin and Receptor Activator of Nuclear Factor-Kappa β Expression

The released form of RANKL and OPG was quantified via an enzyme-linked immunosorbent assay (ELISA) using two commercially available kits (Elabscience^®^, Signal Hill, CA, USA). The cell culture medium was collected at each time point (one, two and three days of daily US stimulation), and 100 μL of each experimental condition was incubated in a 96-well plate pre-treated with the primary antibody specific for the protein of interest. The ELISA assay was performed following the manufacturer’s instructions. The absorbance was read at 450 nm using a microplate reader (Epoch, BioTek, Santa Clara, CA, USA). A standard curve for each protein of interest was prepared, and the amount of RANKL or OPG was calculated accordingly.

### 2.8. Statistical Analysis

Statistical analysis was performed using GraphPad Prism version 6.0 for Windows (Solana Beach, CA, USA). The Shapiro–Wilk test was used to ascertain the data’s normality. For normally distributed data, a one-way ANOVA test was conducted, followed by Tukey’s HSD test for multiple comparisons. For non-normally distributed data, the Kruskal–Wallis test was applied along with Tukey’s HSD test for multiple comparisons. The *p*-values lower than 0.05 were considered statistically significant, and the results are expressed as mean ± standard deviation.

## 3. Results

### 3.1. Initial Optimization by Metabolic Activity

The colorimetric MTT assay was used to assess cellular metabolic activity at the end of 1, 24 and 72 h of single US stimulation.

Concerning the hPLFs, there were no significant differences in the metabolic activity of the stimulated cells compared to the untreated cells after 1 and 24 h (Figure 2A). After 72 h, only the experimental group stimulated with 1.0 MHz at 30 mW/cm^2^ for 5 min induced significant differences when compared to the control (*p* < 0.05). When comparing US parameters, 1.5 MHz at 30 mW/cm^2^ for 5 or 10 min did not alter the hPLFs’ metabolic activity, as opposed to the remaining experimental conditions that statistically increased cell activity throughout the experiment. Overall, the experimental conditions of 1.0 MHz at 30 mW/cm^2^ for 5 min and 1.5 MHz at 60 mW/cm^2^ for 5 min and 10 min appear to be promising for hPLFs stimulation, since they showed superior metabolic activity values in comparison to other stimulation conditions after 24 and 72 h of a single US stimulation session.

The hFOBs also did not show altered metabolic activity 1 h after the single US stimulation (Figure 2B). At 24 h, hFOBs displayed significantly increased metabolic activity after being stimulated with the 1.0 MHz at 30 mW/cm^2^ for 5 min (*p* < 0.001) or 10 min (*p* < 0.01) and with 1.0 MHz at 60 mW/cm^2^ for 5 min (*p* < 0.001) when compared to untreated cells. The effects on the metabolic activity 72 h after stimulation persisted with the experimental conditions of 1.0 MHz at 30 mW/cm^2^ for 5 min (*p* < 0.01) and for 10 min (*p* < 0.05). Other conditions, such as 1.0 MHz at 60 mW/cm^2^ for 10 min (*p* < 0.001), 1.5 MHz at 30 mW/cm^2^ for 10 min (*p* < 0.05) and 1.5 MHz at 60 mW/cm^2^ for 5 min (*p* <  0.001), significantly altered the metabolic activity as well. Overall, the US conditions of 1.0 MHz at 30 mW/cm^2^ for 5 min and 1.0 MHz at 60 mW/cm^2^ for 5 and 10 min seem to be promising, since they induced statistically significant differences in metabolic activity 24 and 72 h after a single US compared to the control and other experimental conditions.

### 3.2. Optimal Parameters Analysis

After the initial optimization, the most promising US parameters were selected based on their ability to enhance cellular metabolic activity. For hPLFs, the most promising conditions were 1.0 MHz at 30 mW/cm^2^ for 5 min and 1.5 MHz at 60 mW/cm^2^ for 5 and 10 min, while for hFOBs, the conditions that resulted in the greatest impact on cell viability were 1.0 MHz at 30 mW/cm^2^ for 5 min and 1.0 MHz at 60 mW/cm^2^ for 5 and 10 min. These cells were stimulated daily for three consecutive days to evaluate the effect of prolonged US stimulation on cell activity and protein expression (i.e., ALP, OPG, RANKL).

#### 3.2.1. Metabolic Activity and Cell Concentration

Concerning the effect of daily US stimulation on the metabolic activity, hPLFs presented significantly increased metabolic activity after one and two days of daily US stimulation with 1.0 MHz at 30 mW/cm^2^ for 5 min (*p* < 0.001) compared to the control (Figure 3A). After three days of stimulation, there were no statistical differences among the conditions. The metabolic activity of hFOBs (Figure 3B) was not altered following daily US stimulation at any of the time points. The US stimulation did not affect the proliferation of hPLFs (Figure 3C) or hFOBs (Figure 3D).

#### 3.2.2. Expression of Osteogenic Markers

The expression of ALP on hPLFs and hFOBs was maintained along the time course of the experiment for all groups, without statistically significant differences compared to the untreated cells (Figure 4A,B).

Concerning the soluble OPG concentration on hPLFs (Figure 4C), the untreated cells showed decreased release of OPG for up to three days. The cells stimulated with 1.0 MHz at 30 mW/cm^2^ for 5 min and 1.5 MHz at 60 mW/cm^2^ for 5 min reverted this tendency, increasing the OPG release on day 3 compared to day 2. However, there were no statistically significant differences between US stimulation and control conditions. Among the stimulation parameters, 1.0 MHz at 30 mW/cm^2^ for 5 min significantly improved the released form of OPG when compared to the experimental conditions, with 1.5 MHz at 60 mW/cm^2^ for 5 min (*p* < 0.01) and 10 min (*p* < 0.05) after three days of stimulation. Also, stimulating with 1.5 MHz at 60 mW/cm^2^ for 10 min induced significantly higher released OPG after two days of stimulation when compared to the same US condition applied for 5 min (*p* < 0.05).

Although there were no statistically significant differences in the OPG expression of hFOBs (Figure 3D), cells exposed to 1.0 MHz at 60 mW/cm^2^ for 5 and 10 min showed a consistent increase in OPG expression along with the number of US sessions. This trend is opposite to the one observed for the control and 1.0 MHz at 30 mW/cm^2^ for 5 min, which exhibited an increased OPG synthesis at day two, followed by a decrease at day three.

To complement this analysis, the released form of RANKL was also quantified using an ELISA assay, but no expression was found; therefore, these data are not shown.

## 4. Discussion

In the present study, we continuously stimulated human osteoblasts and fibroblasts with US and investigated which US parameters were the most promising in modulating their biological response. Our results suggested that the best US parameterization would be 1.0 MHz at 30 mW/cm^2^ for 5 min, 1.0 MHz at 60 mW/cm^2^ for 5 min and 10 min for hFOBs and 1.0 MHz, 30 mW/cm^2^ for 5 min, 1.5 MHz, 60 mW/cm^2^ for 5 min or 10 min for hPLFs, since these conditions significantly and consistently improved their metabolic activity. Also, US maintained the ALP expression and stimulated OPG synthesis, while RANKL expression was not detected, showing that US promoted bone formation events in these cells (Figure 5).

Among the current non-invasive therapies to increase the orthodontic movement, photobiomodulation and vibration are commonly employed due to the commercially available devices that deliver these therapies. However, there are still issues related to their efficacy, paving the way for the search for other potential therapeutic modalities. The application of US in dentistry, although less studied, seems to modulate the biological response of cells involved in OTM. In fact, US stimulation has been reported to stimulate the differentiation and maturation of osteoblasts [25,26] and hPLFs [27,28,29]. Also, previous studies have shown the beneficial effects of US on such cells, promoting their proliferation and activity [20,30,31].

Currently, US is mainly applied at low intensities (e.g., 30 and 60 mW/cm^2^) for 10, 15 or 20 min in pulsed mode. Our study contrasts with the existing studies in the literature, since we intended to investigate the effect of low-intensity US in continuous mode for shorter stimulation times. Our results demonstrate that continuous US stimulation delivered for 5 and 10 min per day is able to positively influence the cells, exposing them to less stimulation. Also, since we applied US at low intensities with shorter durations, we did not expect any temperature increase during stimulation [32,33].

The US stimulation was able to increase the metabolic activity of both hFOBs and hPLFs compared to the control, depending on the US parameters. These findings are in agreement with other studies that demonstrated that US improved metabolic activity [34,35,36,37]. In the initial optimization of US parameters, a single session of US was sufficient to impact the cells’ metabolic activity up to 72 h after the stimulation. Both cell lines responded differently to US stimulation, since the most promising conditions differed between cell types, except for the 1.0 MHz at 30 mW/cm^2^ for 5 min, which was consistently superior to the control in both hFOBs and hPLFs. This suggests that this particular condition could be used to improve osteoblastic and PDL fibroblastic activity.

Following the previous optimization phase, the most promising US parameters were applied daily for three days. The metabolic activity given by the MTS data increased in the first two days after US stimulation and then decreased after the third day for both cell types. This also occurred in the untreated cells, indicating that this cellular response was not caused by the US stimulation. This can be elucidated by analyzing the cell proliferation results. The hPLFs proliferated throughout the study span, which diminished the space available in the culture plate for them to proliferate normally, reducing cellular metabolism on day 3 due to the higher cell confluency. The hFOBs’ behavior was slightly different, since their proliferation significantly increased up to day 2, but decreased at day 3 in both the experimental and control groups. This can be explained because osteoblasts are a cell line and, thus, they have a higher growth rate than hPLFs (primary cells), reaching a higher cell confluency more quickly. Although both cell types reduced their metabolic activity or proliferation, they remained active and continued synthetizing specific osteogenic differentiation markers.

ALP is closely related to the calcification process of mineralized tissues, such as bone [38], and, thus, its quantification is generally used as an early-stage marker of osteogenic differentiation [39]. ALP also works as an activation marker of human osteoblasts following mechanical stimulation [40]. It has been demonstrated that ALP activity in osteoblasts increases from day 5 after daily US stimulation, which can be an indicative of enhanced osteogenesis [22,41]. Also, hPLFs have been reported to synthetize more ALP up to seven days after US stimulation [42,43]. In our study, the ALP expression was constant throughout the experiment for all conditions, with no statistically significant differences being observed between stimulated cells and the control group over three days of daily stimulation. This indicates that further US stimulation sessions are required in order to have a measurable impact on ALP activity.

The OPG is a soluble form of RANKL and acts as its scavenger, thereby inhibiting osteoclast precursors from maturing into osteoclasts. Both RANKL and OPG are synthesized by osteoblasts and fibroblasts [44,45]. The balance between RANKL and OPG expression determines whether bone is formed or resorbed [46,47]. Higher OPG levels result in fewer osteoclasts and, thus, the OPG expression is considered as an early marker for osteoblastic activity and bone formation [48]. Also, it has been reported that, under physiological conditions, hPLFs may synthesize higher levels of OPG compared with RANKL, which has an inhibitory effect on osteoclastogenesis [45]. When the physiological balance is disturbed (e.g., under the influence of mechanical forces), the production of osteoclastogenesis-stimulating molecules is increased, leading to increased osteoclast formation. hFOBs can present increased OPG synthesis following US stimulation, but this response may be dependent on the duration of treatment [49]. Regarding hPLFs, the available literature is scarce in terms of investigating the OPG expression on these cells. Our results showed that certain experimental US conditions appear to have a positive effect on OPG production in both hPLFs and hFOBs. The untreated hPLFs had decreased OPG expression throughout the experiment, while the US stimulation with 1.0 MHz at 30 mW/cm^2^ and 1.5 MHz at 60 mW/cm^2^ for 5 min promoted a recovery of OPG expression after three days. Differently, in the hFOBs, the OPG expression peaked on day 2, followed by a decrease up to day 3. This trend was reversed in the hFOBs stimulated with 1.0 MHz at 60 mW/cm^2^ for 5 and 10 min. These findings suggest that daily US stimulation influences the OPG expression of hPLFs and hFOBs for up to three days, although this biological response was not significantly different from the control. Thus, US should be applied for longer periods to reach statistically significant differences when compared to the untreated cells. In addition, we did not detect RANKL expression in either cell type which indicates that the US stimulation at these parameters does not induce osteoclastogenesis events in either cell, but rather promotes the bone formation, as was confirmed by OPG and ALP expression.

Some limitations of this study should be highlighted. Although we have extensively investigated the effects of US parameters on several biological outcomes (i.e., metabolic activity, cell proliferation, ALP and OPG soluble expressions), other biological analyses should be conducted in the future in order for us to fully understand the impact of such stimulation protocols on cells involved in OTM. Nevertheless, these findings still provide important insight into the US stimulation of hFOBs and hPLFs to accelerate orthodontic movement.

These findings will aid future directions in this field, as our study’s implications and applicability are huge. Although therapeutic US is clinically approved for bone fracture treatment by the Food and Drug Administration (FDA), this therapy is not yet clinically employed as an adjunctive therapy to accelerate OTM [50]. Low-frequency mechanical vibrations (at 30 or 120 Hz) are clinically used to accelerate OTM, but their effects may not be clinically important [51,52,53], thus revealing the need for new adjunctive therapies to effectively accelerate OTM. Ultrasound uses high-frequency mechanical waves (at 1.0 and 1.5 MHz), but the stimulation protocols used in in vitro studies are limited to what is currently approved for bone (i.e., 1.5 MHz, 30 mW/cm^2^, 20 min, pulsed mode), and its cellular effects remain unclear [50]. Therefore, appropriate optimization of US dosage for OTM should be conducted as a starting point before endorsing its use in animal and clinical research. As such, we have investigated the effects of different US parameters on the cells involved in OTM, assessing several biological outcomes, such as cell proliferation, metabolic activity, ALP, OPG and RANKL soluble expression. It is worth noting that we used only human-derived cells, as opposed to previous studies that used mainly animal-derived cell lines, which strengths our results and the potential of this therapy to be applied in orthodontics.

Our findings showed that continuous US, rather than pulsed US, for only 5 min may promote biologic events associated with orthodontic movement, such as bone formation. Employing US in continuous mode for a shorter duration is desirable in order to reduce the treatment time and increase patient adherence. The US at the frequencies proposed, 1.0 and 1.5 MHz, has a penetration depth of up to 5 cm, which assures that the periodontal tissue can effectively absorb the mechanical stimulus [54]. Also, it is a noninvasive and painless therapy, and the parameters reported herein can be easily implemented in clinical settings. As there are already US devices to treat mandibular fractures [55], these can also be employed to specifically target OTM. The success of this therapy in clinical practice will improve orthodontic treatments not only by accelerating the treatment, but also by targeting more complex tooth movements without using invasive techniques. This therapy could also be adjusted and customized according to the clinical condition of each patient, and different stimulation settings could be applied to different teeth, depending on the nature of the OTM. Despite the study´s clinical implications, more in vitro studies are required in order to investigate US for longer stimulation durations and periods to fully understand its effects on osteogenic differentiation markers in long-term application before reaching in vivo and clinical studies.

## 5. Conclusions

Applying US with 1.0 MHz at 30 mW/cm^2^ for 5 min and 1.0 MHz at 60 mW/cm^2^ for 5 min and 10 min for hFOBs, and 1.0 MHz, 30 mW/cm^2^ for 5 min and 1.5 MHz, 60 mW/cm^2^ for 5 min or 10 min for hPLFs, significantly increased the metabolic activity of both osteoblasts and fibroblasts, and positively influenced the OPG synthesis over RANKL, promoting bone formation. Future cell and animal studies should be conducted to better understand the efficacy of US in accelerating orthodontic movement before endorsing its use in clinical practice.

## Figures and Tables

**Figure 1 biomedicines-12-00361-f001:**
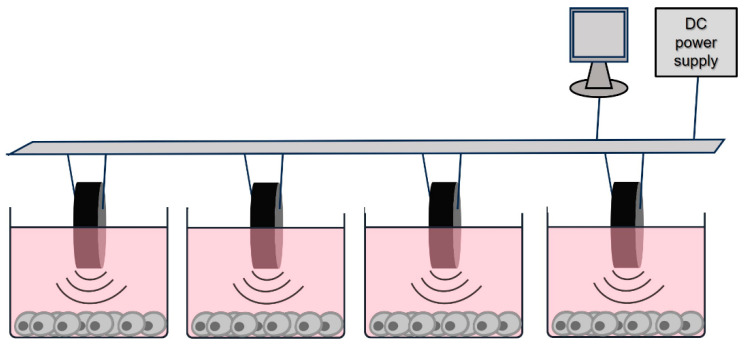
In vitro US stimulation setup.

**Figure 2 biomedicines-12-00361-f002:**
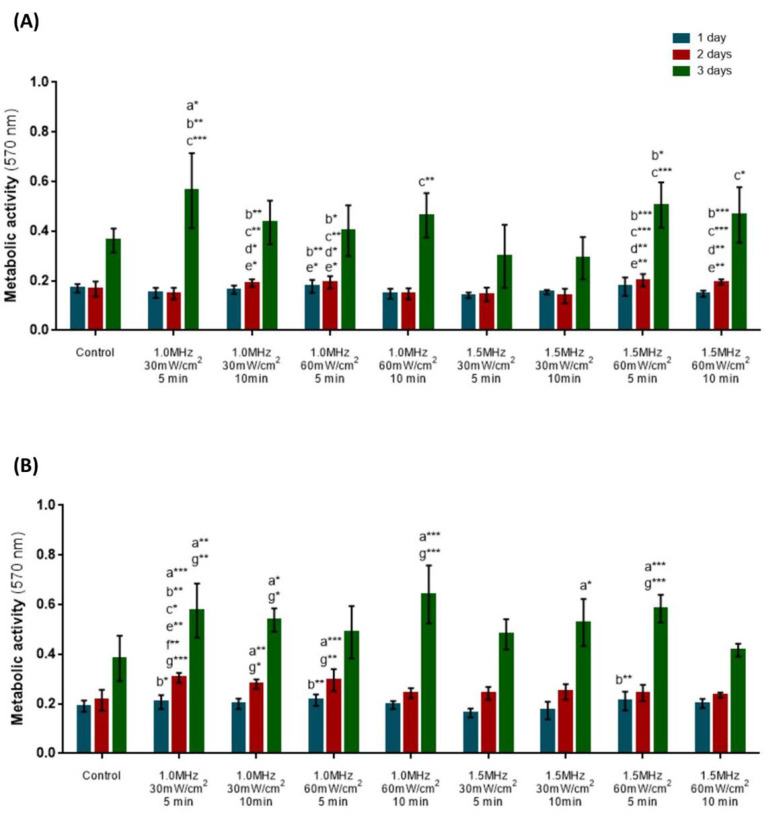
Initial screening of US stimulation parameters through the evaluation of metabolic activity, using the MTT assay, of hPLFs (**A**) and hFOBs (**B**). Statistical differences were detected compared to control: (a) 1.5 MHz, 30 mW/cm^2^, 5 min; (b) 1.5 MHz, 30 mW/cm^2^, 10 min; (c) 1.0 MHz, 30 mW/cm^2^, 5 min; (d) 1.0 MHz, 60 mW/cm^2^, 10 min; (e) 1.5 MHz, 60 mW/cm^2^, 5 min; (f) and 1.5 MHz, 60 mW/cm^2^, 10 min (g). * *p* < 0.05; ** *p* < 0.01; *** *p* < 0.001. Data are represented as mean ± SD (*n* = 4).

**Figure 3 biomedicines-12-00361-f003:**
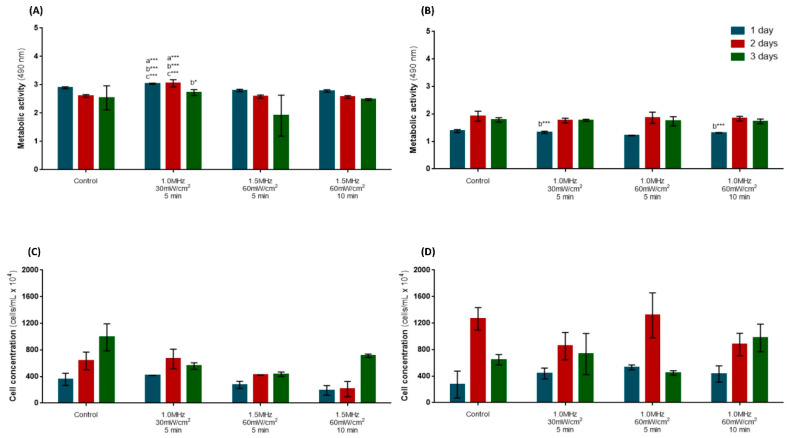
Cellular assays to investigate the optimal parameters for US stimulation. Metabolic activity, assessed through MTS assay, on hPLFs (**A**) and hFOBs (**B**). Cell concentrations of hPLFs (**C**) and hFOBs (**D**). Statistical differences were detected compared to control: (a) 1.5 MHz, 30 mW/cm^2^, 5 min; (b) and 1.5 MHz, 60 mW/cm^2^, 10 min (c). * *p* <  0.05; *** *p* < 0.001. Data are represented as mean ± SD (*n* = 4).

**Figure 4 biomedicines-12-00361-f004:**
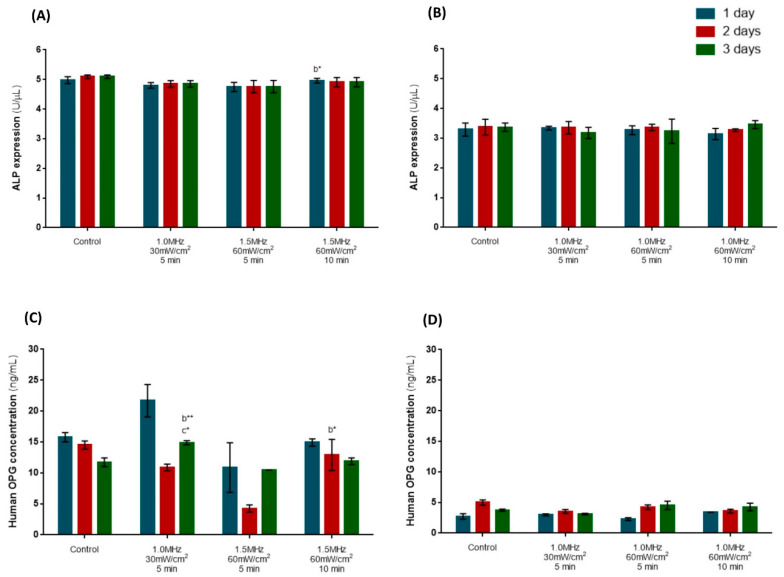
Evaluation of the impact of the optimal parameters for US stimulation on protein expression. The ALP expression on hPLFs (**A**) and hFOBs (**B**). The OPG production by hPLFs (**C**) and hFOBs (**D**). Statistical differences were detected compared to 1.5 MHz, 30 mW/cm^2^, 5 min (b) and 1.5 MHz, 60 mW/cm^2^, 10 min (c). * *p* < 0.05; ** *p* < 0.01. Data are represented as mean ± SD (*n* = 4).

**Figure 5 biomedicines-12-00361-f005:**
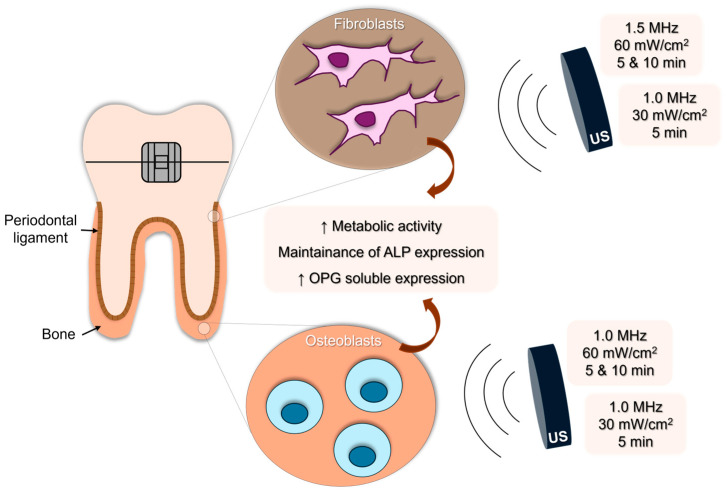
Schematic illustration of the US effects on human osteoblasts and periodontal ligament fibroblasts.

## Data Availability

The data used and analyzed in this study will be available from the corresponding author upon reasonable request.

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
