# Peer review of "Influence of Ultrasound Stimulation on the Viability, Proliferation and Protein Expression of Osteoblasts and Periodontal Ligament Fibroblasts"

_biomedicines, 2024, doi:10.3390/biomedicines12020361_

Round 1
Reviewer 1 Report
Comments and Suggestions for Authors
The paper “Influence of Ultrasound Stimulation on the Viability, Proliferation and Protein Expression of Osteoblasts and Periodontal Ligament Fibroblasts. Running title: Effects of Ultrasound Stimulation on Osteoblasts and Fibroblasts” aimed to clarify the role of US in OTM by exploring different stimulation parameters and their effect on biological response of cells involved in OTM. The article presents sufficient references and meets the readers’ interests of biomedicines. While here are still some shortcomings that need to be further improved or explained.
Comments:
Q1. Are there any data references on the effect of ultrasound intensity on cell growth status? Such as at what intensity, cell growth status is affected.
Q2. The figures in this article lack legend, what groups and treatments do the bars in different colors represent?
Q3. The data analysis in this paper is sufficient, but the amount of data seems to be insufficient, which leads to the limitation to support the conclusion.
Q4. If possible, please supplement the schematic representation of the action mechanism as a figure.
Q5. What are the practical application scenarios of the findings in this paper?
Author Response
REVIEWER 1
The paper “Influence of Ultrasound Stimulation on the Viability, Proliferation and Protein Expression of Osteoblasts and Periodontal Ligament Fibroblasts. Running title: Effects of Ultrasound Stimulation on Osteoblasts and Fibroblasts” aimed to clarify the role of US in OTM by exploring different stimulation parameters and their effect on biological response of cells involved in OTM. The article presents sufficient references and meets the readers’ interests of biomedicines. While here are still some shortcomings that need to be further improved or explained.
REVIEWER COMMENT: Are there any data references on the effect of ultrasound intensity on cell growth status? Such as at what intensity, cell growth status is affected.
AUTHORS RESPONSE: We would like to thank you for your insightful comments and suggestions. The effect of therapeutic ultrasound on growth status is not clear from literature. This is mainly due to the drastically different ranges and not uniform parameters analyzed by the authors. Therefore, it is extremely hard to establish a tendency or a robust range of values. Nonetheless, therapeutic ultrasound is currently approved for the treatment of bone fractures, including mandibular fractures, by the Food and Drug Administration (FDA) at specific parameters, such as 1.5 MHz, 30 mW/cm2 for 20 min, but no information on cellular growth is provided. This underscores the importance in testing a wide range of US parameters in in vitro studies to fully understand its cellular effects.
AUTHORS ACTION: To address this suggestion, we have added some information on the manuscript in Discussion section (page 9):
“Although therapeutic US is clinically approved for bone fractures treatment by Food and Drug Administration (FDA),[50]” (Lines 345-346)
“Ultrasound uses high-frequency mechanical waves (at 1.0 and 1.5 MHz) but the stimu-lation protocols used in in vitro studies are limited to what is currently approved for bone (i.e., 1.5 MHz, 30 mW/cm2, 20 min, pulsed mode),[50] and its cellular effects remain unclear.” (Lines 350-352)
REVIEWER COMMENT: The figures in this article lack legend, what groups and treatments do the bars in different colors represent?
AUTHORS RESPONSE: We apologize for this mistake. Indeed Figure 2 did lack the legend. We would like to inform that each different color represents different timepoints.
AUTHORS ACTION: Please see Figure 2.
REVIEWER COMMENT: The data analysis in this paper is sufficient, but the amount of data seems to be insufficient, which leads to the limitation to support the conclusion.
AUTHORS RESPONSE: We completely understand the authors point of view. In fact, we partially agree with the Reviewer. Even though we amassed an extensive amount of data (e.g., metabolic activity, cell proliferation, ALP, OPG and RANKL soluble expressions) to fully and completely understand the effects of therapeutic ultrasound on osteoblasts and fibroblasts, we consider that we would require even more information. Nevertheless, in our opinion, considerable amount of this information may yet be unattainable through our current available technology, such as complete cell signaling, complete cell metabolism, complete genomic activity, for each cell and for all the cells exposed to the treatment. Thus, we consider that this knowledge, that we thoroughly obtained and analyzed, is relevant for the scientific community and will aid future research on the ultrasound effects on cells involved in OTM.
AUTHORS ACTION: To address this suggestion, we have included some information in the Discussion section (page 9):
“Few limitations of this study should be highlighted. Although we have extensively investigated the effects of US parameters on several biological outcomes (i.e., metabolic activity, cell proliferation, ALP and OPG soluble expressions), other biological analyses should be conducted in the future to fully understand the impact of such stimulation protocols on cells involved in OTM. Nevertheless, these findings still provide an important insight into the US stimulation on hFOBs and hPLFs to accelerate orthodontic movement.” (Lines 338-343)
REVIEWER COMMENT: If possible, please supplement the schematic representation of the action mechanism as a figure.
AUTHORS RESPONSE: Thank you for your comment. We added a new Figure with the schematic representation of the ultrasound mechanisms on osteoblasts and fibroblasts.
AUTHORS ACTION: Please see Figure 5 included in the Discussion section (page 8) along with this sentence:
“Also, US maintained the ALP expression and stimulated OPG synthesis, while RANKL expression was not detected, showing that US promoted bone formation events in these cells (Figure 5).” (Lines 260-262)
REVIEWER COMMENT: What are the practical application scenarios of the findings in this paper?
AUTHORS RESPONSE: Thank you for your comment. To address this, we added information on study´s implications in future research.
AUTHORS ACTION: Please see Discussion section (page 9 and 10):
“Our findings showed that continuous US, rather than pulsed US, for only 5 min may promote the biologic event associated with orthodontic movement, such as bone for-mation. Employing US in continuous mode for shorter duration is desirable to reduce the treatment time and increase patient adherence. The US at the frequencies proposed, 1.0 and 1.5 MHz, has a penetration depth up to 5 cm, which assures that the periodontal tissue can effectively absorb the mechanical stimulus.[54] Also, it is a noninvasive and painless therapy, and the parameters herein reported can be easily implemented in the clinical settings. As there are already US devices to treat mandibular fractures,[55] they can be also employed to specifically target the OTM. The success of this therapy in clinical practice will improve the orthodontic treatments not only by accelerating the treatment duration but also by targeting more complex tooth movements without using invasive techniques. This therapy could be also adjusted and customized according to the clinical condition of each patient and different stimulation settings could be applied to different tooths, depending on the nature of the OTM. Despite the study´s clinical im-plications, more in vitro studies are required to investigate US for longer stimulation times and periods to fully understand its effects on osteogenic differentiation markers in long-term application, before reaching to in vivo and clinical studies.” (Lines 360-376)
Reviewer 2 Report
Comments and Suggestions for Authors
The study aimed to investigate the potential of ultrasound (US) as an adjunctive procedure to accelerate orthodontic tooth movement (OTM). Various stimulation parameters were explored to understand their impact on the biological response of cells involved in OTM, including human fetal osteoblasts and periodontal ligament fibroblasts cell lines. While the research sheds light on the potential of US in this context, there are some critical points and suggestions worth considering:
1. It would be valuable to integrate in vivo or clinical studies to provide a more comprehensive understanding of the practical implications of these findings. Assessing how these parameters affect OTM in real patients would strengthen the study's applicability.
2. Adding a control group would allow for a more robust assessment of the efficacy of US in enhancing OTM.
3. It would be beneficial to explore a wider range of frequencies, power densities, and durations to establish optimal settings for US therapy.
4. Addressing the practical aspects, such as the type of ultrasound devices and protocols that could be employed in orthodontic treatments, would provide valuable insights.
Comments on the Quality of English LanguageMinor editing of English language required.
Author Response
REVIEWER 2
The study aimed to investigate the potential of ultrasound (US) as an adjunctive procedure to accelerate orthodontic tooth movement (OTM). Various stimulation parameters were explored to understand their impact on the biological response of cells involved in OTM, including human fetal osteoblasts and periodontal ligament fibroblasts cell lines. While the research sheds light on the potential of US in this context, there are some critical points and suggestions worth considering:
REVIEWER COMMENT: It would be valuable to integrate in vivo or clinical studies to provide a more comprehensive understanding of the practical implications of these findings. Assessing how these parameters affect OTM in real patients would strengthen the study's applicability.
AUTHORS RESPONSE: We would like to thank you for your time revising our paper and your insightful comments. Ultrasound (US) is a form of mechanical stimulus that has been explored as an alternative to the currently available treatments to accelerate OTM. However, we have identified a significant gap in literature regarding the stimulation protocols that produce the most beneficial effects at cellular level, since there is a scarce of studies assessing the potential of this therapy to accelerate OTM. In fact, to the best of our knowledge, there is no pre- or clinical study published with US stimulation for this purpose. In this sense, our goal was to conduct an extensive optimization of different ultrasound parameters to understand which combination of parameters positively influences the cells´ activity. We selected osteoblasts and fibroblasts from periodontal ligament from human sources as these cells are involved in OTM. Since this study represents a starting point in this field, an in vivo study would not be a feasible option since it would sacrifice a huge number of animals and request substantial resources without knowing a priori which parameters might produce the best biological response. We agree with the Reviewer that animal studies would significantly increase the study´s applicability and, thus, it is our goal to conduct in the future in vivo studies focusing on the application of ultrasound to accelerate OTM. However, we feel that more in vitro studies are needed for the well understanding of the effects of this therapy before endorsing its use in animals and, then, in clinical studies.
AUTHORS ACTION: To address this suggestion, we have acknowledged in the manuscript that future studies are required for the comprehensive understanding the study´s applicability. Please see Discussion section (page 9 and 10):
“Ultrasound uses high-frequency mechanical waves (at 1.0 and 1.5 MHz) but the stimu-lation protocols used in in vitro studies are limited to what is currently approved for bone (i.e., pulsed mode, 30 mW/cm2, 10 to 20 min),[50] and its cellular effects remain unclear. Therefore, appropriate optimization of US dosage for OTM should be conducted as a starting point before endorsing its use in animal and clinical research.” (Lines 350-354)
“Despite the study´s clinical implications, more in vitro studies are required to investigate US for longer stimulation times and periods to fully understand its effects on osteogenic differentiation markers in long-term application, before reaching to in vivo and clinical studies.” (Lines 373-376)
Please see Conclusions section (page 10):
“Applying US with 1.0 MHz at 30 mW/cm2 for 5 min, 1.0 MHz at 60 mW/cm2 for 5 min and 10 min for hFOBs, and 1.0 MHz, 30 mW/cm2 for 5 min, 1.5 MHz, 60 mW/cm2 for 5 min or 10 min for hPLFs significantly increased the metabolic activity of both osteo-blasts and fibroblasts, as well as positively influenced the OPG synthesis over RANKL, promoting bone formation. Future cell and animal studies should be conducted to better understand the US efficacy in accelerating the orthodontic movement, before endorsing its use in clinical practice.” (Lines 379-385)
REVIEWER COMMENT: Adding a control group would allow for a more robust assessment of the efficacy of US in enhancing OTM.
AUTHORS RESPONSE: We would like to thank you for your suggestion. We included a control group in the experimental setting, which consisted of osteoblasts and fibroblasts cultured and handled in the same conditions as the other experimental conditions but without ultrasound stimulation. For the control, we have only placed the stimulation device on top of the well with it turned off.
AUTHORS ACTION: Please see Materials and Methods section (page 3):
“Untreated cells were used as control and operated under the same conditions, but with the US equipment turned off.” (Lines 104-105)
Please see Figures 2, 3 and 4 where the control group is included.
REVIEWER COMMENT: It would be beneficial to explore a wider range of frequencies, power densities, and durations to establish optimal settings for US therapy.
AUTHORS RESPONSE: We would like to thank the Reviewer for the interesting point of view. From literature, we found that ultrasound therapy is currently applied at central frequencies of 1.5 MHz at the intensity of 30 mW/cm2 in pulsed mode for 20 min. In fact, these parameters are clinically approved by Food and Drug Administration for the treatment of bone fractures. Considering what has been done, we have noticed that applying ultrasound in continuous mode, besides pulsed mode, is not explored neither shorter period of time (5 or 10 min), which are desirable to reduce the treatment time and increase patient adherence. Also, besides 30 mW/cm2, we have tested other intensity (i.e., 60 mW/cm2) to understand if higher intensities would produce better results. We have not explored power densities above 60 mW/cm2 because it is well known that, for higher power densities, continuous ultrasound might increase the temperature, which would mask our biological results. Increasing the number of stimulation variables, such as power densities, frequencies and durations, would translate in a wide range of possible combinations of stimulation protocols and, along with the application in two cell types, it would be time-consuming and would demand a wide range of resources. Therefore, we have preferred to focus on potential stimulation parameters that could alter cells´ activity, according to what has been employed in literature. In fact, we have tested eight combinations of ultrasound protocols in each cell type, which is a significant effort in reaching optimal conditions for the stimulation, and to our knowledge is completely inedited. From our study, we observed that from the eight conditions, there were three combinations of protocols that significantly improved the cells activity for each cell type and those combinations were different for both cell types.
REVIEWER COMMENT: Addressing the practical aspects, such as the type of ultrasound devices and protocols that could be employed in orthodontic treatments, would provide valuable insights.
AUTHORS RESPONSE: Thank you for your comment. To address this, we added information on study´s implications in future research.
AUTHORS ACTION: Please see Discussion section (page 9 and 10):
“Our findings showed that continuous US, rather than pulsed US, for only 5 min may promote the biologic event associated with orthodontic movement, such as bone for-mation. Employing US in continuous mode for shorter duration is desirable to reduce the treatment time and increase patient adherence. The US at the frequencies proposed, 1.0 and 1.5 MHz, has a penetration depth up to 5 cm, which assures that the periodontal tissue can effectively absorb the mechanical stimulus.[54] Also, it is a noninvasive and painless therapy, and the parameters herein reported can be easily implemented in the clinical settings. As there are already US devices to treat mandibular fractures,[55] they can be also employed to specifically target the OTM. The success of this therapy in clin-ical practice will improve the orthodontic treatments not only by accelerating the treatment duration but also by targeting more complex tooth movements without using invasive techniques. This therapy could be also adjusted and customized according to the clinical condition of each patient and different stimulation settings could be applied to different tooths, depending on the nature of the OTM.” (Lines 360-373)
Round 2
Reviewer 1 Report
Comments and Suggestions for Authors
No additional problems.